# Anomaly Detection in Fog Computing Architectures Using Custom Tab Transformer for Internet of Things

**Abdullah I. A. Alzahrani** [1], **Amal Al-Rasheed** [2], **Amel Ksibi** [2,*], **Manel Ayadi** [2], **Mashael M. Asiri** [3] **and Mohammed Zakariah** [4]

1. Department of Computer Science, College of Science and Humanity, AlQuwaiiyah, Shaqra University, Riyadh 15526, Saudi Arabia
2. Department of Information Systems, College of Computer and Information Sciences, Princess Nourah bint Abdulrahman University, Riyadh 11671, Saudi Arabia
3. Department of Computer Science, College of Science & Art at Mahayil, King Khalid University, Abha 62529, Saudi Arabia
4. Department of Computer Science, College of Computer and Information Sciences, Riyadh 11442, Saudi Arabia
* Correspondence: amelksibi@pnu.edu.sa

**Abstract:** Devices which are part of the Internet of Things (IoT) have strong connections; they generate and consume data, which necessitates data transfer among various devices. Smart gadgets collect sensitive information, perform critical tasks, make decisions based on indicator information, and connect and interact with one another quickly. Securing this sensitive data is one of the most vital challenges. A Network Intrusion Detection System (IDS) is often used to identify and eliminate malicious packets before they can enter a network. This operation must be done at the fog node because the Internet of Things devices are naturally low-power and do not require significant computational resources. In this same context, we offer a novel intrusion detection model capable of deployment at the fog nodes to detect the undesired traffic towards the IoT devices by leveraging features from the UNSW-NB15 dataset. Before continuing with the training of the models, correlation-based feature extraction is done to weed out the extra information contained within the data. This helps in the development of a model that has a low overall computational load. The Tab transformer model is proposed to perform well on the existing dataset and outperforms the traditional Machine Learning ML models developed as well as the previous efforts made on the same dataset. The Tab transformer model was designed only to be capable of handling continuous data. As a result, the proposed model obtained a performance of 98.35% when it came to classifying normal traffic data from abnormal traffic data. However, the model's performance for predicting attacks involving multiple classes achieved an accuracy of 97.22%. The problem with imbalanced data appears to cause issues with the performance of the underrepresented classes. However, the evaluation results that were given indicated that the proposed model opened new avenues of research on detecting anomalies in fog nodes.

**Keywords:** network security; deep learning; feature selection; intrusion detection

## 1. Introduction

IoT devices are currently widely employed in intelligent applications, including smart cities, healthcare [1], and transportation. All of these IoT-enabled applications share two similar functions: "monitoring" (regularly checking the sensors' state) and "actuating" (acting on the data gathered during monitoring). Additionally, IoT is a networked system built on recognized standards that exchange knowledge. Further, many communication standards, tools, and protocols have been developed due to the many appliance domains. As a result, the Internet of Things (IoT) is frequently referred to as the Internet of People (IoP) because practically everyone uses it regularly, from people to institutions. Moreover,

it enables measurement collection from small, affordable, intelligent end nodes dispersed over a vast physical region with less expensive implementation and operation [2]. However, these advantages come at a cost in terms of finite resources, particularly the end nodes' battery life.

Some of the data collected by IoTs are seen to be unexpected. The shocking data may come from environmental changes, deliberate action, faulty operation, coincidence, or perhaps both. Anomalies were used to describe them [3]. Even though the sensors employed at the edge are insufficient, anomalies are predicted to occur. The end nodes' battery lives may suffer due to the nodes' rapid processing. Due to these restrictions, the network may be more susceptible to errors and malicious attacks [4].

IoT devices are vulnerable to assault since they are connected to the Internet and lack proper security measures. An attacker can swiftly hack IoT devices by taking over smart gadgets that can be used maliciously to exploit other IoT-connected devices [5,6]. Therefore, it is crucial to recognize improper actions to ensure the network operates reliably and securely. Additionally, IoT networks can prevent the broadcasting of useless or inaccurate measurements by spotting intriguing or uncommon events. As a result, the network's dependability can increase while energy consumption is decreased [7].

Anomaly detection entails the identification of noteworthy or unexpected occurrences in the network [8]. Finding a model for the vast majority of normal data is essential to identify anomalies in a dataset. The anomalies can then be identified as those data vectors that considerably depart from the normal model. Finding abnormalities in the network [9] while minimizing overhead and obtaining high detection accuracy is a major challenge.

In the Internet of Things, there are two categories of anomaly detection mechanisms: statistical and machine learning [10]. Only regular IoT traffic is used in statistical methods to create trained models [11]. While doing this, machine learning techniques use both legitimate and malicious communications to train their models. Based on the learning process, these methods are divided into supervised, unsupervised, and semi-supervised categories [12]. During the supervised learning process, the traffic features are mapped to a traffic class, such as normal or assault. Only labeled datasets are used in this learning procedure. By finding intriguing structures in the data, the unsupervised learning process learns the traffic features without being aware of the traffic class. Unsupervised learning groups comparable data in semi-supervised learning, whereas labeled data is used to categorize unlabeled data.

The current detection methods for anomalies depend primarily on a centralized cloud's [13] inability to address IoT requirements, such as resource allocation and scalability. With IoT, operations are carried out across many devices, and large amounts of data are exponentially generated [14]. Since it enables users to access Internet-based services, the cloud is essential to the Internet of Things (IoT). However, because of its centralized architecture, it cannot manage IoT devices even while it does expensive calculations. The great distance between an IoT device and the centralized anomaly detection system also results in a high detection time. Since the centralized cloud environment can accommodate the service requirements of IoT, anomaly detection in IoT differs from currently used methodologies [15]. A brand-new distributed intelligence technique called "computation" is used to reduce the gap. The fog exchanges information by processing data near the data sources, i.e., IoT devices. At the fog layer, as depicted in Figure 1, where fog nodes perform dispersed processing, security measures can be put into place [16]. To implement distributed security mechanisms, expensive computations and storage from IoT devices may be offloaded [17].

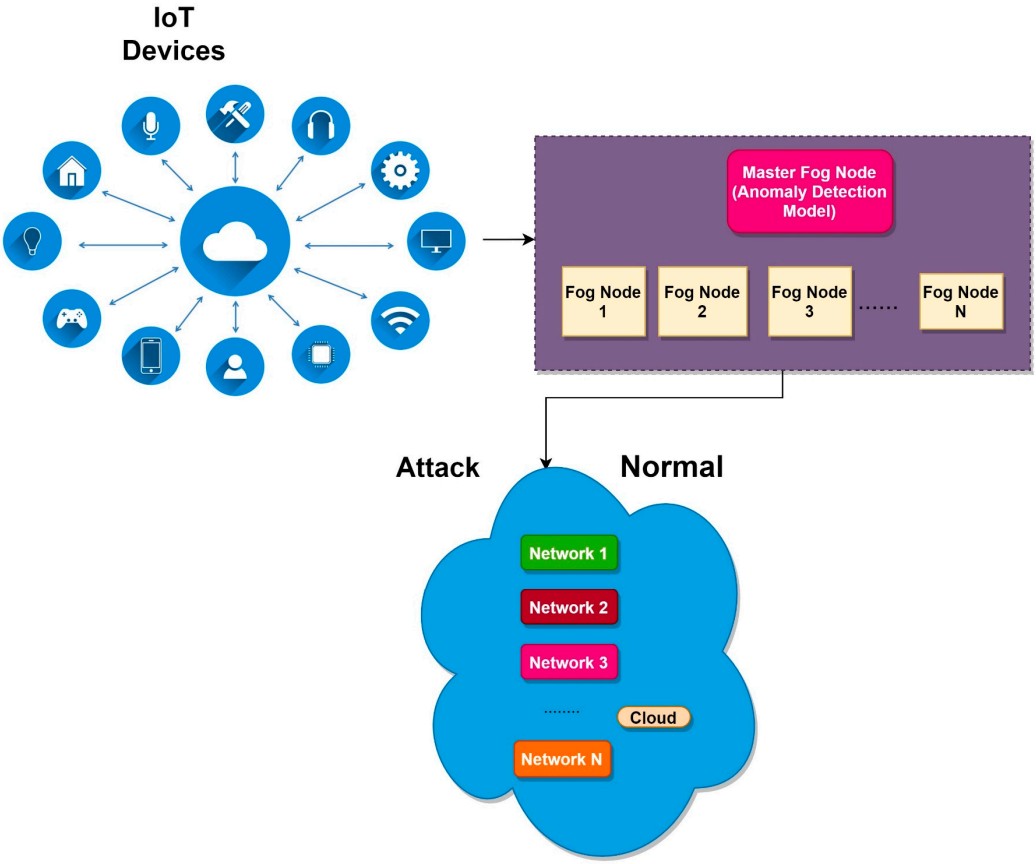

**Figure 1.** Implementation of the anomaly detection system in IoT systems.

In this study, we introduced a framework model and a hybrid algorithm for effective ML algorithm selection to discover workable methods for anomalies and incursion IoT network traffi in a fog environment from many ML algorithms.

Significant contributions of the current work include:

1.  The majority of Intrusion Detection System (IDS) related works are based on the outdated KDD cup99 or NSL-KDD [3,4,11,18–21] data sets, which do not include the majority of contemporary assaults. In contrast to the KDD cup99 dataset, we employed the latest data set (UNSW-NB15) that covers the most recent attacks in this study.
2.  Comparative study of the performance of the traditional ML models in anomaly detection
3.  A modified Tab transformer model is proposed. It is the first time this unique technology has been used to detect fog node anomalies.

The rest of the paper's organization is as follows. Section 2 discusses the literature review of the past work done. The proposed strategy, including data sets, model construction, and performance evaluations, is discussed in Section 3. In Section 4, we use experiments to evaluate the proposed methodologies quantitatively. Then, we discuss the methods and results, conclude the paper in Section 4 and finally conclude in Section 5.

## 2. Literature Review

This section presents relevant research and comprehensive background information on machine learning (ML) selection for detecting anomalies and intrusions in IoT networks [22] traffic.

Anomaly detection in IoT data using deep learning was proposed by [15], and it was shown to be more effective than a conventional IDS for identifying coordinated IoT Fog assaults. The NSL-KDD intrusion dataset was used. Compared to the standard model's

binary classification recall of 97.50%, it achieved a score of 99.27% for deep learning. In addition, machine learning earned an average recall of 93.66% in multi-classification, whereas deep learning scored an average recall of 96.5%.

The authors of [23] suggested cognitive fog computing for IDS in an IoT network. The suggested methodology could detect malicious behavior in nearby fog nodes as opposed to employing a centralized cloud-based infrastructure. The cloud stores a list of all fog nodes for future research. The proposed model is assessed using the NSL-KDD dataset, and detection is accomplished using the online sequential extreme learning machine (OSELM) method. Their model has a 0.37% FAR and a 97.36% accuracy rate.

The authors of [24] suggested an adaptive IDS for IoT that can recognize DoS threats. In this work, a fresh dataset was gathered using Wireshark over the course of four consecutive days on an IoT testbed. Unfortunately, their suggested model outperforms the Naive Bayes classifier.

The authors of [25] proposed an IDS based on neural networks and locust swarm optimization. For this experiment, which makes use of the NSL-KDD and UNSW-NB15 datasets, the accuracy and FAR are 94.04 and 2.21%, respectively.

Li et al. suggested a combined K-means clustering technique with a PCA fog computing design for anomaly detection. An ELM-based Semi-supervised Fuzzy C-Means (ESFCM) technique was put out by [12]. The NSL-KDD dataset was utilized. The suggested system outperformed the centralized attack detection framework in terms of performance. It reported an accuracy rate of 86.53% and a decreased detection time of 11 milliseconds.

To put in place an adaptive Intrusion Detection System (IDS) that can recognize when a Fog node has been hacked, and then take the appropriate action to ensure communication availability [26], authors in [18] developed an Anomaly Behavior Analysis Methodology based on Artificial Neural Networks [27] and ensemble approach [21]. The training dataset was produced using the IoT testbed. The accuracy rate of the approach was 97.51%.

Similarly, the authors in [22] suggested a variational long short-term memory (VLSTM) learning model based on reconstructed feature representation for intelligent anomaly identification. Experiments using the publicly available UNSW-NB15 IBD dataset demonstrate that the proposed VLSTM model can successfully address the imbalance and high dimensional issues, and that it can also significantly improve accuracy and decrease false rates in anomaly detection.

By dividing the Intrusion Detection System functions across the fog nodes and the cloud, [19] low resource overheads are achieved. As a result, an accuracy of up to 98.8% was achieved. In addition, compared to installing a neural network on the fog node a 10% decrease in the energy usage of the fog node is observed.

This work is novel since it develops intrusion detection for IoT traffic using SDN and deep learning. SDN enables intelligent network management by separating the control and data planes. In the current IDS, deep learning-based classifiers outperform traditional classifiers in terms of results. Any infiltration in networking systems, in particular IoT networks, is detected by the suggested model [28]. Current existing work related to Anomaly detection is listed in Table 1.

Encryption is necessary to safeguard and stop such errors in transmitting delicate data over the internet and other networks. To strengthen the safety of the delicate data or information, the author created an improved variety of the Caesar cipher in this paper and developed a technique in which flexible arithmetic is used to transform plaintext into ciphertext. The author also created a decryption method that is entirely unrelated to encryption by incorporating divisibility tests and arithmetic modulo.

**Table 1.** Existing work on Anomaly detection.

| Reference | Technique | Dataset | Results |
|---|---|---|---|
| [15] | deep learning | NSL-KDD | 96.5% |
| [23] | online sequential extreme learning machine (OSELM) | NSL-KDD | 97.36% accuracy rate |
| [25] | neural networks and locust swarm optimization | NSL-KDD and UNSW-NB15 | 94.04 |
| [9] | K-means clustering technique with a PCA | NSL-KDD | 86.53% |
| [18] | Artificial Neural Networks | IoT testbed | 97.51% |
| [22] | variational long short-term memory (VLSTM) | UNSW-NB15 IBD | improve accuracy and decrease false rates |
| [19] | LSTM | UNSW-NB15 IBD | 98.8% |

The conventional approach to situational awareness prediction in network security is comparatively simple. For perception and prediction, only one algorithm is typically utilized, and its prediction accuracy is constrained. This study optimizes a radial basis function (RBF) neural network using the simulated annealing (SA) algorithm and the hybrid hierarchy genetic algorithm (HHGA). Hence, it constructs an RBF neural network prediction model based on the HHGA optimization and performs relevant experiments to investigate the application impact of intelligent learning algorithms. The results show that the projected scenario value of the enhanced RBF neural network is relatively close to the actual situation value in 15 instances. The neural network has a significant predictive influence and can assist with network security maintenance [29].

To the best of our knowledge, no study demonstrates which ML algorithm is efficient for the identification of IoT dangerous traffic, despite numerous research proposals on various identification models for accurately detecting IoT malicious traffic. Most academics conduct experiments to evaluate the ML algorithm's performance and, based on the results, they choose the most efficient method. However, it is crucial to research and find the most efficient machine learning method for anomaly and intrusion in IoT network traffic identification by reviewing frequently cited and primarily studied literature reviews.

## 3. Materials and Methods

### 3.1. Dataset

The current analysis employed the UNSW-NB15 dataset as a benchmark [30–32] Previous datasets, including NSLKDD [33], KDD98, KDDCUP 99 [34], CIDDS-001, DARPA, and ADFA were already accessible for Network Intrusion Detection System (NIDS) research [35]. These datasets, most of which date back more than 20 years, have several limitations, making them unreliable and out-of-date. Such datasets are no longer thought to provide a complete or accurate representation of contemporary attack environments, and algorithms trained on such datasets will not exhibit realistic output performance. These databases distort regular traffic and exclude modern attack types, making it simple for stealthy/spy attacks to pass for normal activity.

The following dataset-specific issues also exist: n uneven number of records from various types of traffic, an excessive number of attacks, incomplete training sets that do not accurately reflect all attacks found in the testing set, a dearth of validation work, data generation techniques, and low data rates, etc. [36,37].

The Australian Center for Cyber Security (ACCS) produced a more recent dataset in collaboration with several specialists worldwide to solve the problems presented by earlier datasets in the field. It has been a publicly available dataset for the current NIDS since 2015. As indicated in Table 2, the dataset has 45 total network attributes, including flow and network-based properties. Flow, fundamental, substance, time, and other created features are additional classifications. Approximately 2.5 million CSV-formatted records in total, including 175,341 training data and 82,331 testing data, constitute the entire dataset. The training and testing datasets are devoid of duplicate data to guarantee NIDS evaluation

dependability. Two distinct traffic labels are initially applied to the dataset (attack and normal). The attack categories in Table 2 are further classified into nine more class types according to the attack type.

**Table 2.** Type of attacks present in the UNSW-NB15 dataset and their description.

| Attack Type | Description |
| --- | --- |
| Normal | The transaction that occurs naturally without any threat |
| Generic | A collision attack approach that uses hash functions to conflict the block-cipher configuration. |
| Exploits | A set of instructions (software code) frequently included in malware that exploits faults or security vulnerabilities caused by unintended network behavior. |
| DoS | An attempt to stop genuine users from using web services by flooding the network or server with erroneous login requests and causing it to malfunction or stall |
| Fuzzers | Inputting several permutations and combinations of the data automatically into a "target programme" until one of the combinations uncovers a vulnerability. |
| Reconnaissance | The gathering of pertinent information about a target network or server to evade network security mechanisms. |
| Analysis | An assault designed to infiltrate web applications via port scanning, spam emails, and HTML file penetration. |
| Worms | It is a collection of self-replicating viruses that consumes traffic bandwidth. |
| Backdoor | An effort to circumvent standard authentication methods for remote database and file server access. |

### 3.2. Data Preprocessing

In Machine Learning, more data results in more accurate models. However, data from the real world is inconsistent, noisy, incomplete, and consists of missing values as it is compiled utilizing data mining and storage. Therefore, it is crucial to pre-process raw data into the processed form. The data preparation enhances data quality so that valuable insights can be extracted. This will be beneficial for model development and training. The approaches used to pre-process the UNSW-NB15 dataset are described below.

### 3.3. Data Cleaning

We tried to list the count of the missing values in the dataset corresponding to each feature. The feature "service" had 94,168 missing values for the train set and 47,153 for the test set. After removing the records with missing features, the count of the records corresponding to each class in the total dataset has been reduced. Figure 2 shows the modified distribution of categories in the total dataset.

### 3.4. Data Transformation

The characteristics "proto", "service", "state", and "attack cat" contained categorical information that could not be directly put into the ML models. We utilized "One-hot-encoding" to encode absolute values into the binary format, except for "attack cat," which was the target multiclass attack label that the model had to predict. The columns of the three one-hot encoded characteristics were eliminated, bringing the total number of classes to 61.

The range of the numerical characteristics in the dataset is varied. Therefore, it was essential to normalize the values. Except for the "id" and "label" columns, the numerical feature columns have been normalized using the "MinMaxScaler."

For binary categorization of the characteristics into "normal" and "abnormal", the "labels" column was encoded using LabelEncoder() as "0" for the normal class and "1" for the abnormal class. Again, the binary dataset contains 61 columns.

For multiclass classification, the "'attack cat' attribute's nine categories were label encoded as 0 ('Analysis'), 1 ('Backdoor'), 2 ('DoS'), 3('Exploits'), 4('Fuzzers'), 5('Generic'), 6('Normal'), 7('Reconnaissance'), and 8 ('Worms'). Consequently, the total number of attributes in the multiclass classification dataset has increased to 69.

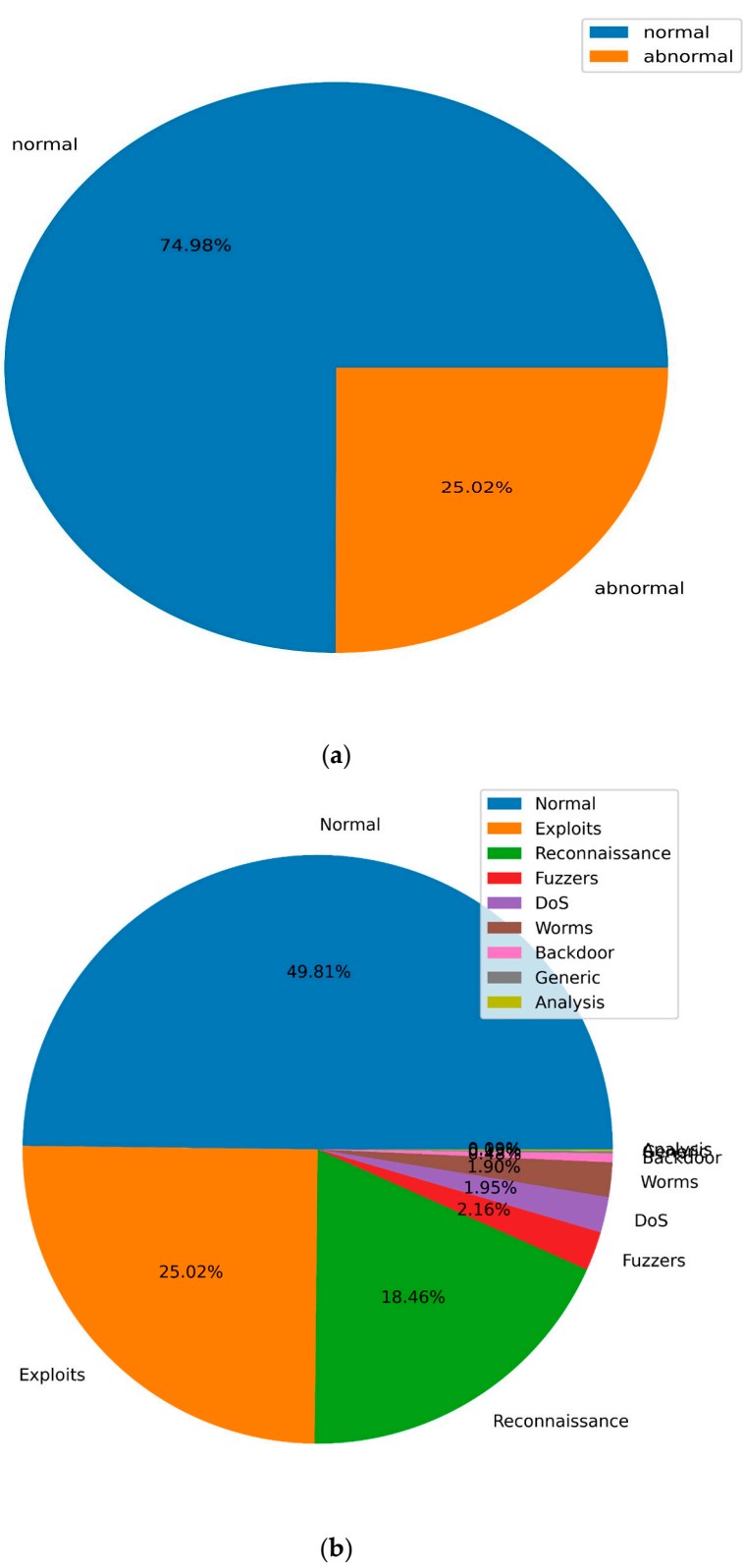

(**a**)

(**b**)

**Figure 2.** The distribution of labels (**a**). Binary (train) (**b**). Multiclass (train).

*3.5. Feature Selection*

Feature selection is essential for the efficient training of machine-learning models [38]. This is because the selection of the features contributes the most to accomplishing a task and eliminates unneeded or redundant qualities [39]; otherwise, the model can learn from noise

and collect insignificant patterns. Consequently, feature selection enhances processing and prediction reliability [38]. In this paper, correlation-based feature selection is used.

### 3.6. Model Development

3.6.1. Tab Transformer

The widely used Transformer design by the authors in [40] served as an inspiration for the TabTransformer architecture that was developed by the authors in [41]. A column embedding layer, a stack of N Transformer layers, and a multilayer perceptron are the components of the suggested design [42]. As described by [43], each Transformer layer comprises a position-wise feed-forward layer, followed by a multi-head self-attention layer. In the study that we are currently presenting, we have utilized a variation from the modified tab transformer model that was proposed by the authors in [44]. The proposed model is illustrated in Figure 3. The revised version only utilized the Tab-transformer's capability to handle the continuous input features. It removed the categorical features and the subsequent normalization layer and concatenation layer related to these features. In other words, it only used the Tab-transformer to handle the continuous features in the input.

The detailed methodology for detecting anomalies in the fog node is depicted in Figure 4.

3.6.2. Model Training Pipeline

Following the data cleaning process, there were 1,41,321 data samples. In total, 80% of those samples were designated for training, while the remaining 20% were used for testing. The sklearn and keras libraries were utilized during the development of the machine learning models. Pytorch-widedeep is responsible for the implementation of the Tab transformer. A total of ten epochs were used to train the tab transformer model. On the NVidia T4 GPU with 40 GB of RAM, it took 15 s for each epoch to complete.

3.6.3. Performance Evaluation

Accuracy: The ratio of the number of correct predictions to the total number of predictions represents how often the classifier makes accurate predictions, as shown in Equation (1).

Recall: The fraction of true positives successfully identified, as shown in Equation (2)

Precision: Proportion of anticipated positives that are positive, as shown in Equation (3)

F1 score: The harmonic mean of recall and precision, as shown in Equation (4).

$$\text{Accuracy} = (TP + TN)/(TP + TN + FP + FN) \tag{1}$$

$$\text{Recall} = TP/(TP + FN) \tag{2}$$

$$\text{Precision} = TP/(TP + FP) \tag{3}$$

$$\text{F1 score} = (2 \times \text{Precision} \times \text{Recall})/(\text{Precision} + \text{Recall}) \tag{4}$$

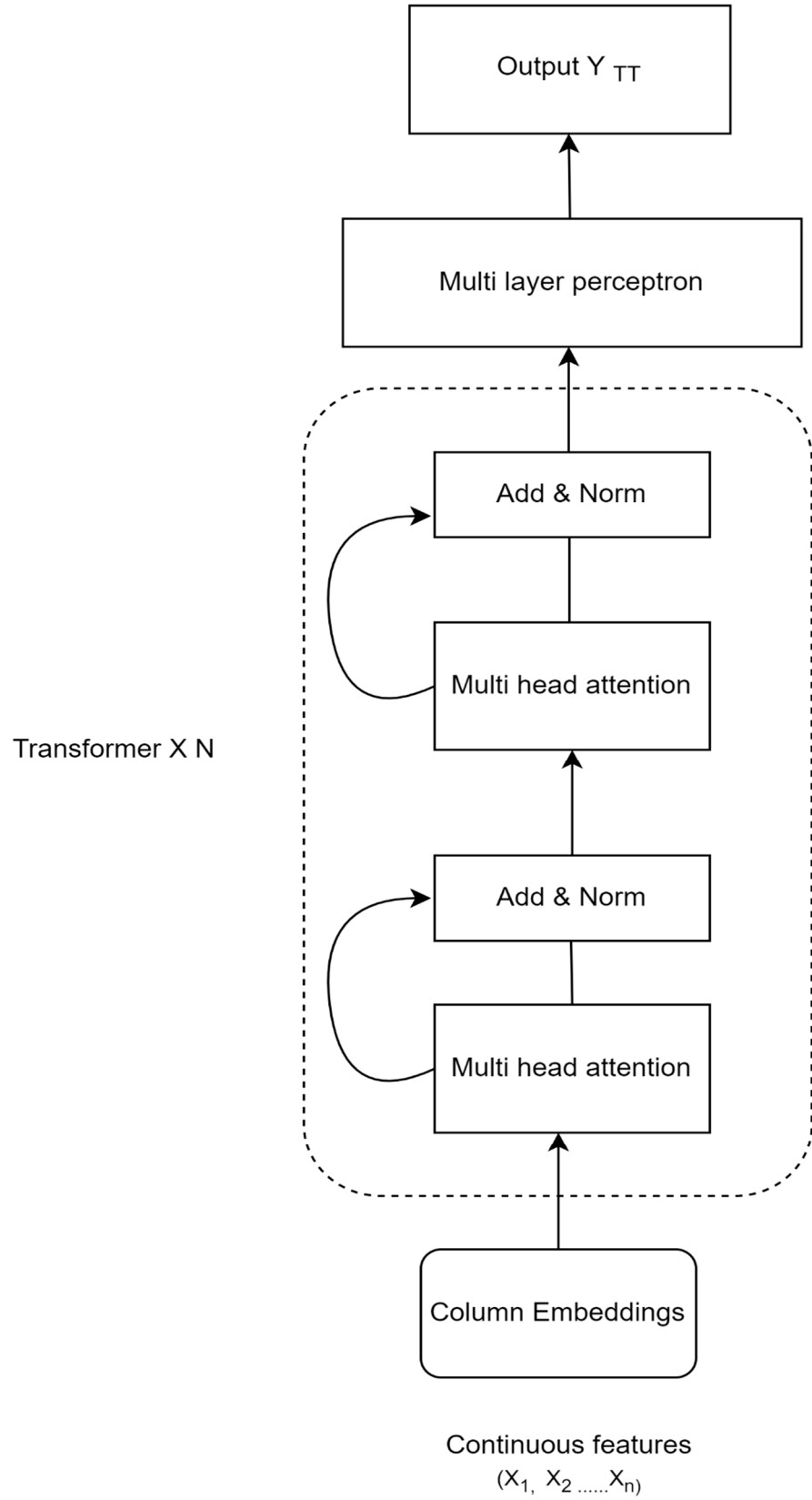

**Figure 3.** Proposed tab transformer architecture.

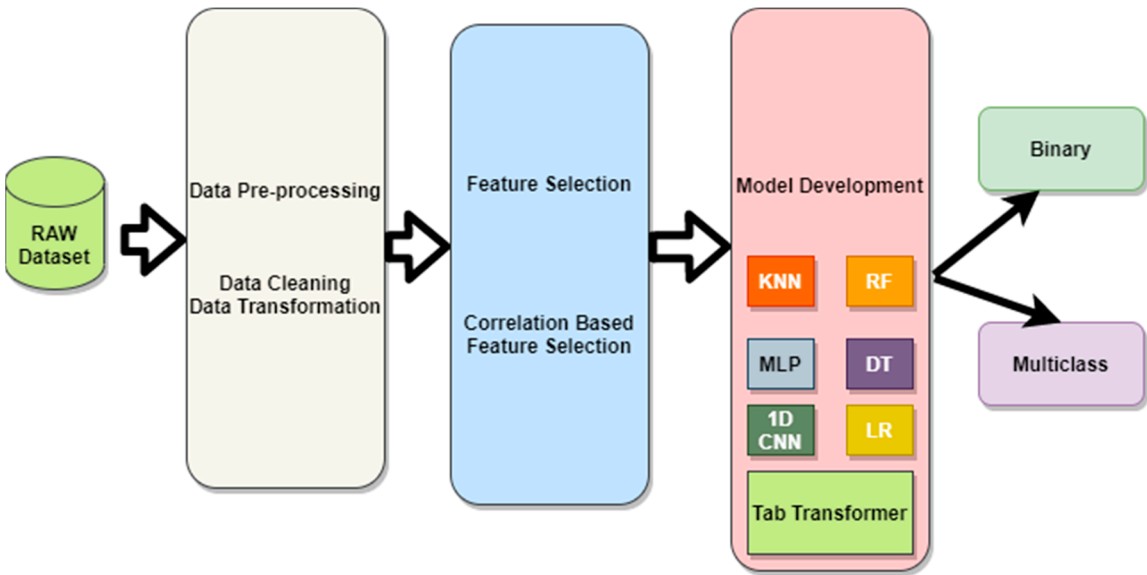

**Figure 4.** Proposed framework for detecting anomalies from IoT networks.

## 4. Experimental Environment and Discussion

### 4.1. Feature Selection

The correlation matrix of features in the binary dataset, excluding the ID feature, is displayed in Figure 5. This graph illustrates the correlational relationships between dataset features. Using this correlation matrix graph, the importance of the features could be understood. By establishing a threshold for the correlation with the label feature, we extracted only the most essential features from the binary dataset with 61 features.

### 4.2. Multi Class Dataset

The Correlation matrix of features in the multiclass dataset is shown in Figure 6. From the correlation matrix, if a threshold of 0.3 is set up with the label class 14, features were shortlisted from the 69 feature original multiclass dataset.

### 4.3. Model Development

This section presents the results of the numerous machine-learning algorithms used for the chosen features. The results of the many classic machine learning models applied to the binary classification (normal/abnormal problem) are provided in Table 3.

**Table 3.** Performance of the binary classification (normal/abnormal) of the data by various ML models.

| Classifier | Accuracy |
|---|---|
| Random Forest Classifier | 96% |
| Decision Tree Classifier | 96.5% |
| K Nearest Neighbor Classifier | 95.72% |
| Linear Support Vector Machine | 95.96% |
| Logistic regression | 95.88% |
| MLP Classifier | 95.88% |
| 1D CNN | 96.8% |

The 1D CNN performs significantly better than other classical ML models when classifying the records into normal or abnormal categories. However, the performance of the other models was not significantly lower. There is only a marginal discernible change in performance. The results of the various machine learning models' attempts to categorize

the data into multiple attack categories are presented in Table 4. Here also, the highest performance was achieved was for 1D CNN.

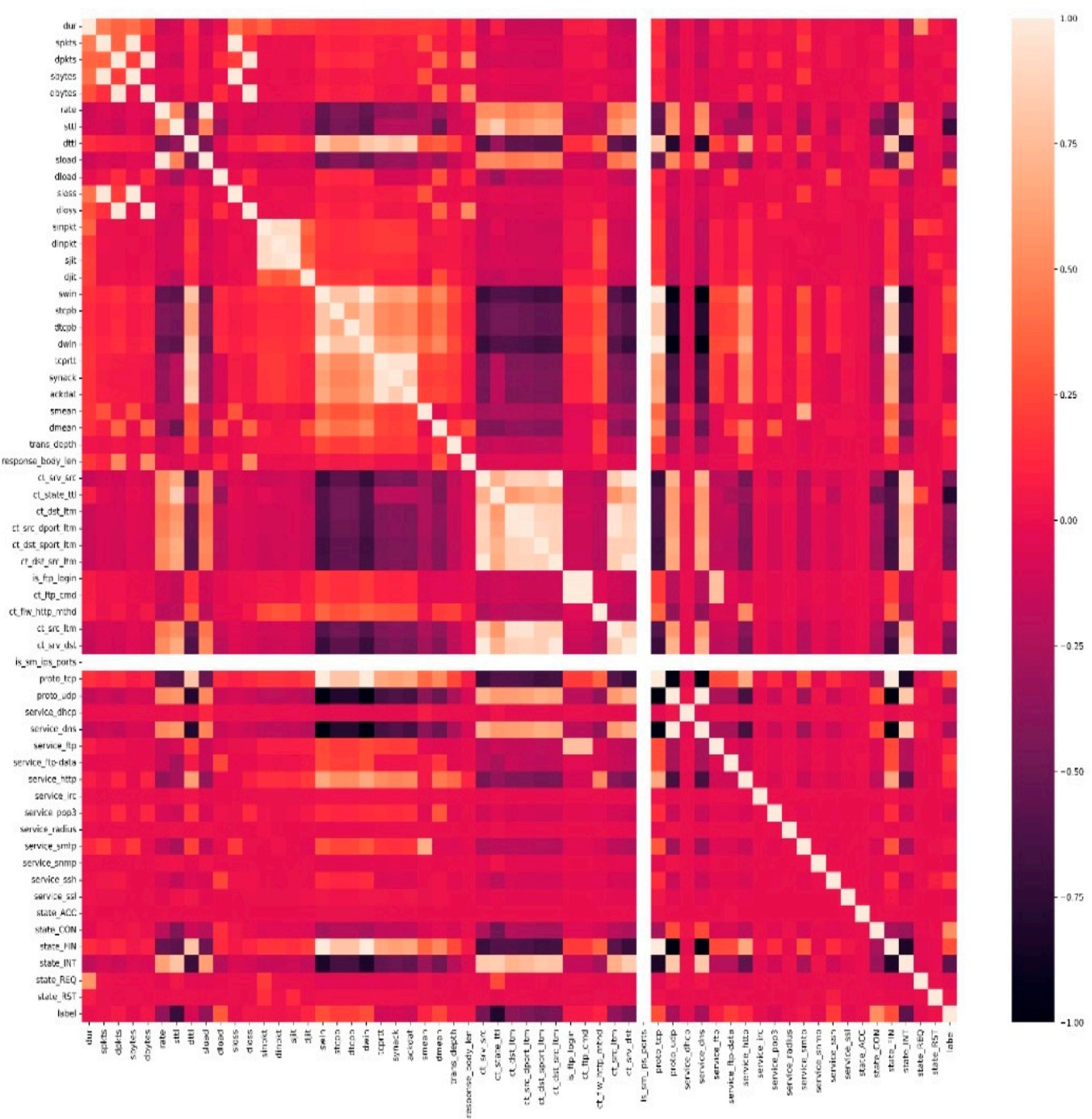

**Figure 5.** Correlation matrix of the binary class dataset. By setting a threshold value of >0.3, 15 columns of features from the total of 61 columns are extracted.

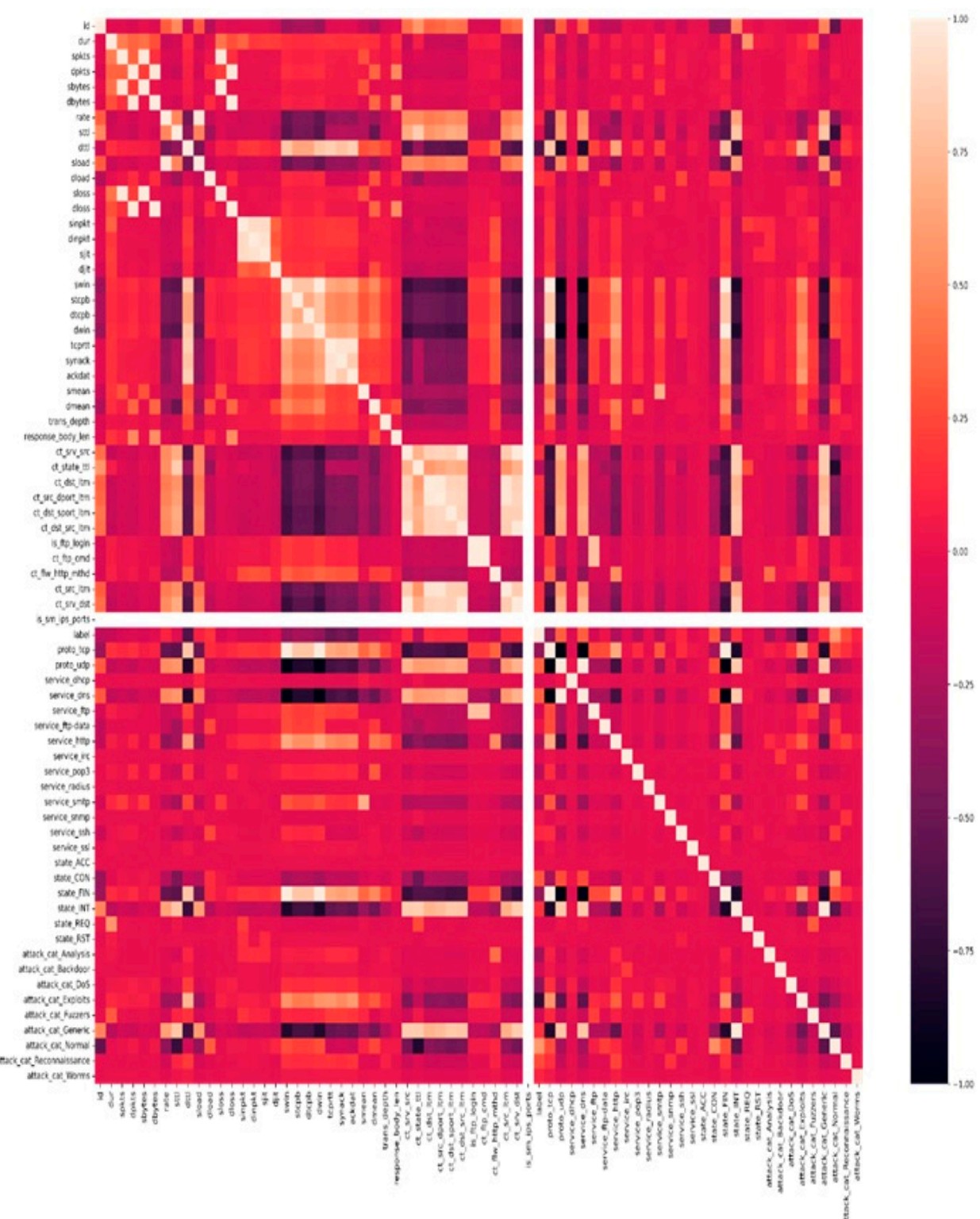

**Figure 6.** Correlation matrix of the multiclass dataset.

**Table 4.** Performance of the multiclass classification of the data by various ML models.

| Classifier | Accuracy |
|---|---|
| Random Forest Classifier | 95.43% |
| Decision Tree Classifier | 95.5% |
| K Nearest Neighbor Classifier | 96% |
| Linear Support Vector Machine | 95.01% |
| Logistic regression | 94.96% |
| MLP Classifier | 94.96% |
| 1D CNN | 96.23% |

### 4.3.1. Performance of the Customized Tab Transformer

A performance evaluation of the tab transformer was carried out after it had been subjected to training for 10 iterations. The metrics accuracy and loss plots for the training and validation datasets showed that the model performed adequately on both the training dataset and the validation dataset when the training was being carried out. This indicated that the model was fit for use. On the other hand, loss plots displayed a steady decline in quality, in contrast to the train's accuracy and the test set's results, which both showed persistent signs of improvement. The plots in Figures 7 and 8 revealed that the model did not clearly demonstrate any evidence of overfitting. This was likely because the feature selection handled the risk of learning unnecessary information, which may cause the model to learn noise data and cause overfitting.

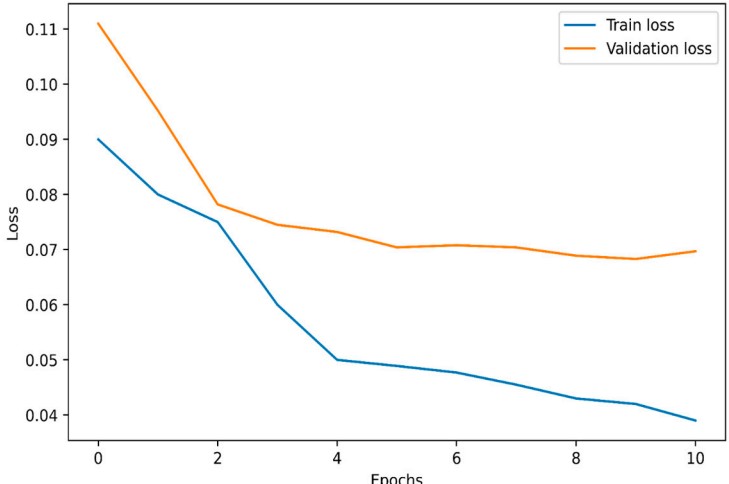

**Figure 7.** Loss plot for binary classification (Normal vs. Abnormal).

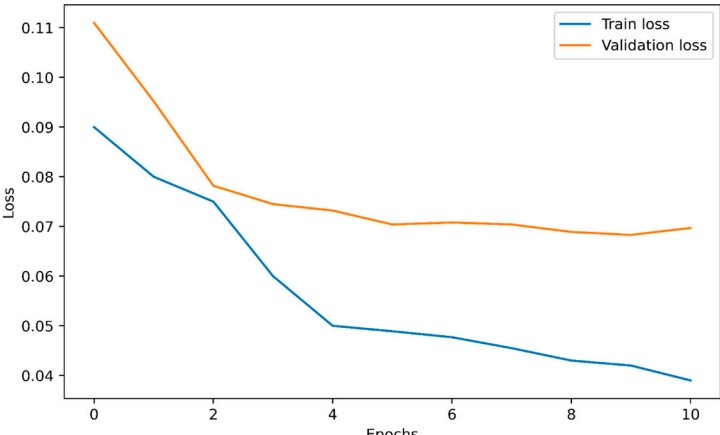

**Figure 8.** Loss plot for multiclass classification (9 classes).

The performance of the suggested tab transformer for the binary and multi-class problems is provided in Table 5. According to the findings, the newly generated model has a better performance than the typical machine learning models that have already been produced, as shown in Tables 3 and 4. In addition, the findings suggested that the Tab transformer model was only trained for a limited amount of epochs, yet despite this, it was still able to demonstrate satisfactory performance.

**Table 5.** Performance evaluation of the tab transformer.

| Model | Accuracy | Recall | Precision | F1 Score |
|---|---|---|---|---|
| Tab transformer (binary) | 98.35% | 95.79% | 97.60% | 96.68% |
| Tab transformer (multiclass) | 97.22% | 57.37% | 55.02% | 56.04% |

However, in addition, an essential piece of information was gleaned from the findings. In the instance of the binary classification, every single metric of evaluation was granted a satisfactory performance. Even the problem of uneven data distribution between the normal and abnormal classes does not significantly contribute to the model's performance with regard to the classification of the groups (Figure 9a), as Table 6 demonstrates.

**Table 6.** Class wise performance evaluation of the tab transformer for binary classification.

| Classes | Precision | Recall | F1 Score | Support |
|---|---|---|---|---|
| Abnormal | 99% | 99% | 99% | 17,450 |
| Normal | 98% | 96% | 97% | 5821 |

However, the accuracy metrics were the only ones that showed a positive result in terms of multiclass classification. The remaining metrics hovered around the 50% mark. This may be because of a problem with the data imbalance that occurred in the initial dataset. Table 7 also reveals that certain classes, such as "Analysis," "Backdoor," and "Worms," which have a smaller number of instances of representation in the training dataset, exhibit an almost null value in terms of their precision, recall, and accuracy as they were confused with the over-represented classes (Figure 9b).

**Table 7.** Class wise performance evaluation of the tab transformer for multiclass classification.

| Classes | Precision | Recall | F1 Score | Support |
|---|---|---|---|---|
| Analysis | 0% | 0% | 0% | 105 |
| Backdoor | 0% | 0% | 0% | 22 |
| DoS | 100% | 100% | 100% | 530 |
| Exploits | 100% | 100% | 100% | 4375 |
| Fuzzers | 44% | 48% | 46% | 425 |
| Generic | 99% | 99% | 99% | 11,506 |
| Normal | 100% | 100% | 100% | 5821 |
| Reconnaissance | 52% | 70% | 59% | 463 |
| Worms | 0% | 0% | 0% | 24 |

### 4.3.2. Discussion

Safety concerns were elevated since the systems that are based on the Internet of Things are advancing at a rapid rate. Any action taken by an Internet of Things device that was not intended to be taken could result in significant damage; hence, these devices need to be carefully monitored [45]. However, on the cloud side, it is incredibly challenging to solve the problem since a large volume of data is arriving at this end. Hence, a more effective strategy would be the identification of unexpected patterns of data, also known as an anomaly, on the side of the fog node. As a result, the security procedures can be implemented at the fog layer, which is where fog nodes are located when distributed

processing is being carried out. As a result, the burden of performing costly computations and storing data on IoT devices might be offloaded. This would enable the deployment of distributed security mechanisms. The primary objective of this design should be to reduce the number of false positive alert instances while improving its detection accuracy.

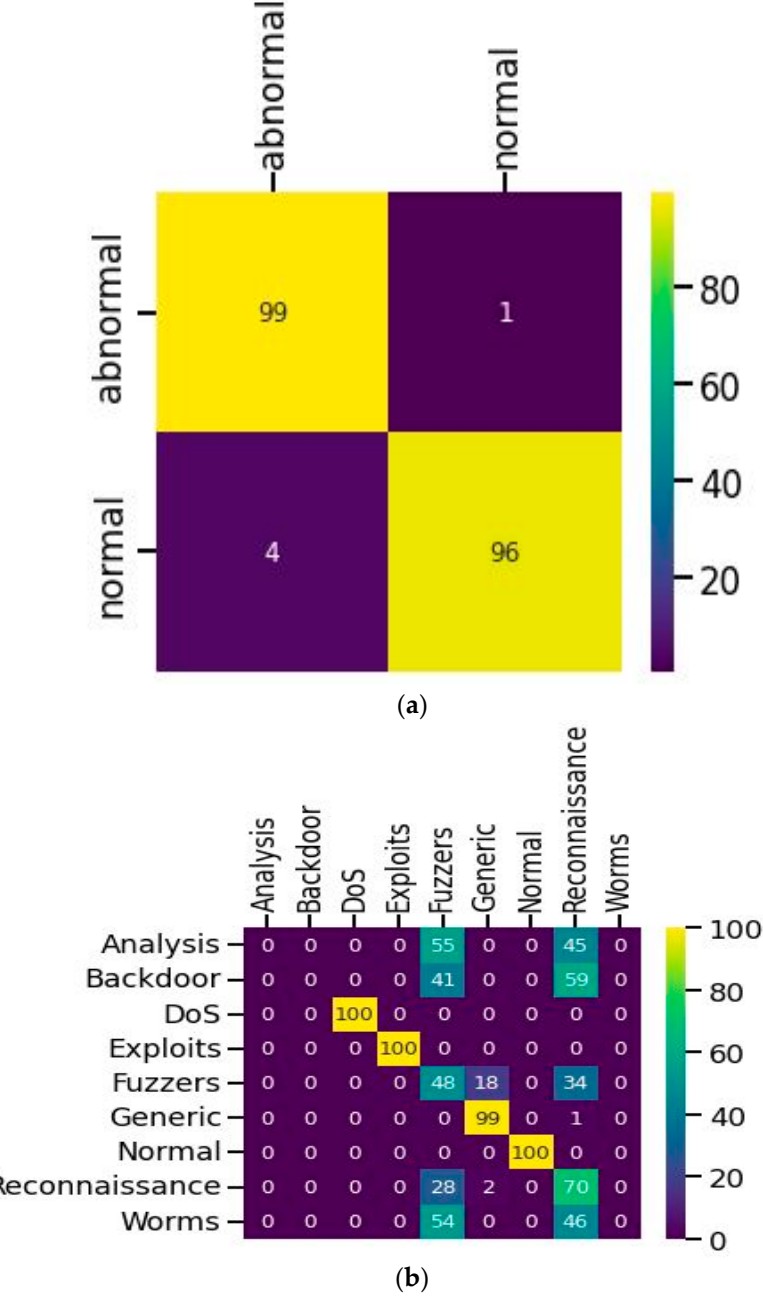

**Figure 9.** Confusion metrics of the (**a**) Binary and (**b**) Multiclass anomaly detection.

Even though several methods already exist for anomaly detection in the IoT devices, such as firewalls and rule-based methods systems [44], these methods appear insufficient to detect unknown attacks. This is likely because these methods cannot keep up with the most recent and sophisticated attacks, which are most researched and have different patterns mostly unknown to the rule book. In addition, it is incredibly challenging to write rules for each of these attacks.

Therefore, the most effective way would be ML-based methods, as they enable systems to learn and improve through the use of historical data. ML-based computer programs do

not require explicit engineering (programmed). They are capable of acquiring knowledge on their own [46]. In the vast majority of prior research for identifying abnormalities at the fog node, conventional ML techniques were utilized and yielded the best results. The authors of [47] employed conventional ML techniques to identify assaults at the edge nodes with the highest accuracy. As demonstrated in Tables 3 and 4, we obtained the best performance for classical ML models in our study. Before the classification task, feature selection has been performed in the two studies. In contrast to [47] which utilized Chi-Square which is a filter-based feature selection strategy, we have employed correlation-based methods. As a result, the feature selection technique undoubtedly contributed to the high accuracy, as it eliminates noise from the data. However, the most significant contribution to the fog node-based anomaly detection model is that it will be lightweight. Since it handles a smaller number of features, allowing it to be readily integrated into fog nodes, which are battery-constrained and computationally less intelligent.

The model created in this paper is a tab transformer that outperforms existing machine learning (ML) approaches and is a novel way of studying anomaly detection in the UNSW-NB15 dataset, as shown in Table 5. We use this technique since it has been demonstrated to be superior to contemporary deep neural networks for tabular data [44] and aim to manage data imbalance issues more intelligently. Given that our study comprises both binary and multi-classification and that the dataset at hand is uneven in terms of the number of entries (Figure 4), accuracy may not be an adequate performance evaluation criterion. Precision, recall, and F1 scores must be considered while comparing various ML algorithms [48]. The binary classification results (normal/abnormal) demonstrated that the Tab transformer model proposed successfully resolved the data imbalance issue, as shown in Table 6.

On the other hand, for multiclass classification, the model achieved high accuracy but not the highest performance in other metrics, as shown in Table 5. However, the performance by class revealed a correlation between the performance of the class prediction and the quantity of class samples, as shown in Table 7. Prior to training a tab transformer, a data augmentation strategy or more data collection for underrepresented classes would be preferable. Clearly, balancing the dataset may generate a model with Tab transformer architecture that performs better.

Even though the data imbalance affected the multiclass categorization of anomalies, the suggested model appears to beat many of the earlier attempts in the domain using the same dataset. Despite the complexity of the feature selection procedure, authors in [49] were able to reach an accuracy of 90.85% for binary classification using decision tree architecture. The decision tree model built in our work got 96.5% accuracy with correlation-based feature selection. Support Vector Machine was created by authors in [48] to detect binary and multiclass abnormalities in the dataset. However, the model's accuracy was poor. As the original dataset is unbalanced between classes, the work suggested that accuracy may not be the most appropriate performance metric for evaluating the model's performance. Authors in [50] developed an integrated rule-based approach that differs from machine learning (ML) approaches. The proposed work is truly innovative as they developed rules for understanding the features from multiple classes. However, they were only able to achieve an accuracy of 84.83% on multiclass prediction, as shown in Table 8. These values are taken from the respective papers.

It would appear that the proposed model is a more suitable fit for the identification of abnormalities at the fog node. On the other hand, a few items need to be addressed shortly, such asthe problem of data imbalance when performing multiclass classification tasks. In addition, the fact that edge devices typically have limited computation resources and memory, necessitates that we carry out an exhaustive analysis of the amount of computation power and the amount of time necessary for predicting anomalies, given that the model must be deployed at the fog node.

**Table 8.** Comparison of the proposed model with the previously developed anomaly/intrusion detection systems.

| Ref. | Models Used | Dataset | Accuracy (Binary Classification) | Accuracy (Multi Class Classification) |
|---|---|---|---|---|
| [19] | LSTM | UNSW-NB15 IBD | 98.8% | - |
| [25] | Neural networks and locust swarm optimization | NSL-KDD and UNSW-NB15 | 94.04% | - |
| [22] | Variational long short-term memory (VLSTM) | UNSW-NB15 IBD | improve accuracy and decrease false rates | - |
| [49] | XGBoost-based feature selection + Decision tree | UNSW-NB15 | 90.85% | - |
| [48] | Support Vector Machine | UNSW-NB15 | 82.11% | 86.04% |
| [50] | Integrated rule-based model | UNSW-NB15 | - | 84.83% |
| **Proposed model** | **Customized Tab transformer** | UNSW-NB15 | **98.35%** | **97.22%** |

## 5. Conclusions

The current work proposed a fog-based anomaly detection system for IoT networks. Implementation of anomaly detection indicated that fog nodes can be utilised effectively in decentralizing an IoT-based network based on cloud architecture. The suggested model was developed on the UNSW-NB15 dataset and employed its architecture to identify aberrant traffic in IoT networks. The proposed detection technique reduced the number of features for multiclass and binary datasets using correlation-based feature selection. However, the test dataset remains unbalanced. Yet, both the ML and suggested Tab transformers demonstrated satisfactory performance. Our Tab transformer design outperforms conventional ML models and obtained 98.35% accuracy on binary classification (Normal vs. Abnormal Traffic) and 97.22% accuracy on multiclass detection jobs.

Furthermore, by comparing the performance of the proposed model to that of previously created models on the same dataset, we have proved the significance of the correlation-based feature selection method. As IoT devices have varying memory capacity, network bandwidth, and battery life limits, we might construct a lightweight anomaly detection model by utilising an optimum collection of attributes. In the future, we intend to test the performance of the proposed model utilising additional balanced IoT-based data sets and conduct performance research of the proposed model in terms of computation complexity and time.

*Limitations*

Although the proposed methodology to detect anomalies is performing better than others, it nevertheless has some limitations.

The computational complexity can be further increased by applying new data augmentation techniques. Further, the parameters can be reduced by applying customized models. A few more features can be added to enhance the accuracy further.

The techniques used by authors in [19] are lightweight and human immune. Whereas we applied the Tab transformer technique in our work, we offer a novel intrusion detection model capable of deployment at the fog nodes to detect the undesired traffic towards the IoT devices by leveraging features from the UNSW-NB15 dataset. A further limitation of the study [19] is that it did not give a comparison with other works. Furthermore, no technical details are clearly mentioned about the features extracted and how the processing was done.

**Author Contributions:** A.I.A.A. drafted the problem and designed the methodology for implementation, A.A.-R. worked on dataset exploration, A.K. started the implementation beginning with data preprocessing, M.A. implanted the designed plan, M.M.A. drafted the initial paper and designed the sections, M.Z. finished the paper writing aby compiling all the sections together with formatting and English proofreading. All authors have read and agreed to the published version of the manuscript.

**Funding:** This study is supported by Princess Nourah bint Abdulrahman University Researchers Supporting Project number (PNURSP2022R235), Princess Nourah bint Abdulrahman University, Riyadh, Saudi Arabia.

**Data Availability Statement:** The dataset used in this work is taken from: N. Moustafa and J. Slay, "UNSW-NB15: a comprehensive data set for network intrusion detection systems (UNSW-NB15 network data set)," in 2015 military communications and information systems conference (MilCIS), 2015, pp. 1–6.

**Conflicts of Interest:** The authors declare no conflict of interest.

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
