# Peer review of "Anomaly Detection in Fog Computing Architectures Using Custom Tab Transformer for Internet of Things"

_electronics, doi:10.3390/electronics11234017_

Round 1

Reviewer 1 Report (Previous Reviewer 3)

1. The Introduction section still needs to be improved.

2. The authors must follow the previous comment to improve the Related Works section.

3. It is not clear what is the novelty of this work. The authors can include step-wise discussion in the proposed scheme.

4. Delete section 7.

5. Mention the limitation of this work in section 6.

6. Sections 4 and 5 can be merged.

7. Improve the discussion about results.

8. English Language should be improved significantly.

Author Response

Comment 1. The Introduction section still needs to be improved.

Author Response: The introduction section is further improved by incorporating a few more contents.

Comment 2. The authors must follow the previous comment to improve the Related Works section.

Author Response: The following papers are not added in the reference list

  1. Deep image retrieval using artificial neural network interpolation and indexing based on similarity measurement
  2. Hyperspectral anomaly detection via memory-augmented autoencoders
  3. A Novel Hybrid Encryption Method to Secure Healthcare Data in IoT-enabled Healthcare Infrastructure
  4. A novel technique for accelerating live migration in cloud computing
  5. Maintainable stochastic communication network reliability within tolerable packet error rate
  6. Multi-Authority CP-ABE-Based Access Control Model for IoT-Enabled Healthcare Infrastructure
  7. Achieving a decentralized and secure cab sharing system using blockchain technology

Comment 3. Delete section 7.

Author Response: This section is now deleted.

Comment 4. Mention the limitation of this work in section 6.

Author Response: Limitations are added.

Limitations: Although the proposed methodology to detect Anomaly is performing better than others but still it has few limitations:

The computational complexity can be further increased by applying new data augmentation techniques. Further, the parameters can be reduced by applying customized models. Few more features can be added to enhance the accuracy further.

Comment 5. Sections 4 and 5 can be merged.

Author Response: Section 4 and 5 are merged.

Comment 6. Improve the discussion about results.

Author Response: The discussion section is improved by discussing more on results.

Comment 7. English Language should be improved significantly.

Author Response: The paper is again checked thoroughly for English language by a native English speaker.

Reviewer 2 Report (Previous Reviewer 2)

1. Why "The Tab transformer model proposed seems to perform well on the existing dataset and outperforms the traditional Machine Learning ML models developed and the previous efforts made on the same dataset."

Why it is still "seems to". 

2. Past published work on Anomaly detection
Rename it with existing work

3. Image quality is still poor

4. Have you reimplemented the Table 7 existing work? If yes how or mentioned clearly you took the results directly from papers.

5. What was the criteria of selection of benchmark papers?

6. In Table 7, the datasets of [37,38,39] are different from yours selected one UNSW-NB15. How can you directly compare accuracy of them with your proposed method?

7. Shift table 1 in your result section and discuss the direct comparison with those technique using UNSW-NB15 dataset.

Author Response

Anomaly Detection in Fog Computing Architectures Using Custom Tab Transformer for Internet of Things

Reviewer 2:

Comments and Suggestions for Authors

Comment 1. Why "The Tab transformer model proposed seems to perform well on the existing dataset and outperforms the traditional Machine Learning ML models developed and the previous efforts made on the same dataset."

Why it is still "seems to".

Author Response: Corrected. Thank you so much.

Comment 2. Past published work on Anomaly detection

Rename it with existing work

Author Response: Corrected. Thank you so much.

Comment 3. Image quality is still poor

Author Response: Images are enlarged now it is clear.

Comment 4. Have you reimplemented the Table 7 existing work? If yes how or mentioned clearly you took the results directly from papers.

Author Response: This table is not implemented. The results were taken from the respective papers.

Comment 5. What was the criteria of selection of benchmark papers?

Author Response: The criteria for selecting the benchmark papers were, Dataset, Methodology and year of publication.

Comment 6. In Table 7, the datasets of [37,38,39] are different from yours selected one UNSW-NB15. How can you directly compare accuracy of them with your proposed method?

Author Response:  All the comparison is done with similar dataset UNSW-NB15

Ref

Models used

Dataset

Accuracy (binary classification)

Accuracy (Multi class classification)

[24]

neural networks and locust swarm optimization

NSL-KDD and UNSW-NB15

94.04

-

[21]

variational long short-term memory (VLSTM)

UNSW-NB15 IBD

improve accuracy and decrease false rates

-

[46]

XGBoost-based feature selection + Decision tree

UNSW-NB15

90.85%

-

[45]

Support Vector Machine

UNSW-NB15

82.11%

86.04%

[47]

Integrated rule-based model

UNSW-NB15

-

84.83%

Proposed model

Customized Tab transformer

UNSW-NB15

98.35%

97.22%

Comment 7. Shift table 1 in your result section and discuss the direct comparison with those technique using UNSW-NB15 dataset.

Author Response: Thank you for the suggestion. All the relevant fields from Table 1 are inserted in the final table of comparison.

Reviewer 3 Report (Previous Reviewer 1)

The authors have addressed all comments from the previous review.

However, I still feel some images must be corrected/improved. For instance, Figure 3 has very low quality. Check if this is occurring during the generation of the PDF file or if it is a problem with the source image.

Author Response

Anomaly Detection in Fog Computing Architectures Using Custom Tab Transformer for Internet of Things

Reviewer 3

Comment: The authors have addressed all comments from the previous review.

Author Response: Thank you very much.

However, I still feel some images must be corrected/improved. For instance, Figure 3 has very low quality. Check if this is occurring during the generation of the PDF file or if it is a problem with the source image.

Author Response: It has now been enhanced.

Round 2

Reviewer 1 Report (Previous Reviewer 3)

This paper can be accepted for publication. 

Author Response

Anomaly Detection in Fog Computing Architectures Using Custom Tab Transformer for Internet of Things

The authors would like to thank the reviewers for their valuable comments. We have incorporated all the comments and suggestions in the revised version of the paper.

Reviewer 1

This paper can be accepted for publication. 

Author Response: Thank you very much

Reviewer 2 Report (Previous Reviewer 2)

1- Current images quality is still poor. Please check after pdf version. e.g. Fig 1 and 2.

2- The response of selection of benchmark papers is still poor. That must be  align with your current study/problem. I still suggest to add relevant papers.

"Author Response: The criteria for selecting the benchmark papers were, Dataset, Methodology and year of publication."

3- Author should add the Ref[18] in table 8 and clearly mention the limitations of Ref[18] to distinguish or highlight the proposed work.

4- English should be improved.

Author Response

Anomaly Detection in Fog Computing Architectures Using Custom Tab Transformer for Internet of Things

The authors would like to thank the reviewers for their valuable comments. We have incorporated all the comments and suggestions in the revised version of the paper.

Reviewer 2

Comment 1: Current images quality is still poor. Please check after pdf version. e.g. Fig 1 and 2.

Author Response: The images are now improved in PDF version. In a word, it is clearer, but while converting to PDF it is fading, but we enhanced the DPI of the source image from 96DPI to 300 DPI and applied various operations to have a good quality image in PDF.

Comment 2: The response of selection of benchmark papers is still poor. That must be  align with your current study/problem. I still suggest to add relevant papers.

Author Response: The selection of the benchmark papers where based on the problem we addressed in the paper. Following are new papers cited in the paper.  

  • Bustamante-Bello, Rogelio, Alec García-Barba, Luis A. Arce-Saenz, Luis A. Curiel-Ramirez, Javier Izquierdo-Reyes, and Ricardo A. Ramirez-Mendoza. "Visualizing Street Pavement Anomalies through Fog Computing V2I Networks and Machine Learning." Sensors22, no. 2 (2022): 456.
  • Labiod, Yasmine, Abdelaziz Amara Korba, and Nacira Ghoualmi. "Fog Computing-Based Intrusion Detection Architecture to Protect IoT Networks." Wireless Personal Communications(2022): 1-29.
  • de Souza, Cristiano Antonio, Carlos Becker Westphall, and Renato Bobsin Machado. "Two-step ensemble approach for intrusion detection and identification in IoT and fog computing environments." Computers & Electrical Engineering98 (2022): 107694.

Comment 3: Author should add the Ref[18] in table 8 and clearly mention the limitations of Ref[18] to distinguish or highlight the proposed work.

Author Response: The techniques used in this work [19] is lightweight and human immune. Whereas we applied the Tab transformer technique. In our work, we offer a novel intrusion detection model capable of deployment at the fog nodes to detect the undesired traffic towards the IoT devices by leveraging features from the UNSW-NB15 dataset. A further limitation of [19] is that it did not give a comparison with other works. Also, no technical details are mentioned clearly about the features extracted and how the processing was done.

Comment 4: English should be improved.

Author Response: Now the paper is thoroughly checked for English Proofreading by a native English speaker.  

This manuscript is a resubmission of an earlier submission. The following is a list of the peer review reports and author responses from that submission.

Round 1

Reviewer 1 Report

The authors present an interesting research topic and very relevant nowadays. However, some improvements are necessary in terms of format and style:

- Most of the images and figures have very low resolution, which does not allow understanding them completely. The font used in them is very small (e.g. fig 5 and 6) and the height-width ratio is lost in some of them. Some figures are too small with a lot of information (fig 1) and some others are just the opposite (fig 4). 

- Figures 7a and 8a, in addition to the format problems explained before, have mistakes in the y-axis (loss instead of resolution).

- When using initials, such as IoT, ML and IDS, they must be explained in their first appearance and used extensively in the rest of the document. However, IoT is used in the first sentence of the introduction without its definition (abstract is normally not considered). IDS in page 3 is never defined.

- There are isolated (or orphan) sentences in the document, e.g. line 68. On the other hand, there are paragraphs that are too long and do not have a proper structure, e.g. lines 199-222.

- Tables must be greatly improved. Some of them have a proper presentation (table 1), but most of them are not using an adequate structure and are mixed with the the rest of the text (tables 2-7).

- Some sections are missing an introductory paragraph (e.g. 4.1 and 4.1.1). In the case of sub subsection 4.1.1 it does not make sense to have a division 4.1.1 if there is not going to be at least a 4.1.2.

- In terms of the English language and style, some sections are very well organized and presented (e.g. section 1), but some others have many typos, misplaces capital letters (e.g. line 332) and composition problems (e.g. section 4).

- Sections are poorly organized. For instance, section 4 is repeated as Results and Discussion.

In terms of the methodology and the proposed algorithm, some important points are missing:

- What was the setup (software, platform, environment, etc.) utilized to developed this system? This information should be included in the Materiales and Methods section.

- If this is oriented to fog computing, how the limitations of an embedded device are considered as part of the design of the algorithm? How would this be evaluated? Isn't a deployment on an embedded device necessary to validate this proposal?

- A better justification and explanation of the Tab Transformer is necessary.

- The performance metrics are poorly presented and justified.

- The feature selection, which is paramount for the algorithm, is also poorly presented and justified. 

Reviewer 2 Report

In this study, author targeted to proposed an anomaly detection system for IoT networks. Intention was to effectively detect the anomaly in fog nodes that further be utilized in IoT-based network. Results seems acceptable, but not compared with recent work. Meanwhile, writing and presentation quality is too poor. Need extensive improvements.

Reviewer 3 Report

1. Please improve the Abstract.

2. The Introduction section is very poor. In a research article, the introduction section must be very strong with the motivations of this paper, which is missing in this paper. Moreover, the disadvantages of the existing schemes must be discussed to motivate this new work.

3. The Literature Review section is poor. The authors must include some more schemes. Also, the following papers must be cited to improve this section, as well as the Reference section:

a) SDN-based intrusion detection system for IoT using deep learning classifier (IDSIoT-SDL)

b) Revisiting shift cipher technique for amplified data security

c) Introduction to the special section on advances of machine learning in cybersecurity (VSI-mlsec)

d) Boosting image watermarking authenticity spreading secrecy from counting-based secret-sharing

e) DNACDS: Cloud IoE big data security and accessing scheme based on DNA cryptography

f) Research on internet security situation awareness prediction technology based on improved RBF neural network algorithm

g) An efficient and time saving web service based android application

4. In section 2, a table can be given to summarize the entire section.

5. What is the use of data cleaning?

6. In section 3.4, how data transformation happening? 

7. Figure 4 is completely not clear.

8. Search stage is completely unclear.

9. Which entities are involved for key management? 

10. In section 4, add the “Experimental Environment” section.

11. What is the source of dataset? Whether it is authentic or not? Mention clearly.

12. How the results of Figure 5 are generated?  

13. Technical details about results are missing.

14. How the training is done?

15. What is the novelty of this work? It is hard to identify from the current version of this paper.

16. Use a well-known software to draw the diagrams of the results section. Some diagrams are blurred.

17. The organization of the paper must be improved. The paper must be formatted properly.

18. Improve the English language.

19. The Reference section must be improved significantly.